# The High Prevalence of Sarcopenia in Rheumatoid Arthritis in the Korean Population: A Nationwide Cross-Sectional Study

**DOI:** 10.3390/healthcare11101401

**Published:** 2023-05-12

**Authors:** Dongwoo Kim, Yu Jin Lee, Eui Seop Song, Ahreum Kim, Cho Hee Bang, Jae Hyun Jung

**Affiliations:** 1Department of Medicine, Korea University College of Medicine, Seoul 02841, Republic of Korea; blustick@hanmail.net (D.K.);; 2Department of Internal Medicine, Korea University Ansan Hospital, Ansan 15355, Republic of Korea; 3Department of Medicine, CHA University School of Medicine, Seongnam 13496, Republic of Korea; 4Department of Nursing, College of Health Science, Honam University, Gwangju 62399, Republic of Korea; choice-bch@hanmail.net

**Keywords:** rheumatoid arthritis, sarcopenia, prevalence

## Abstract

Rheumatoid arthritis (RA) includes musculoskeletal symptoms that lead to disuse atrophy of muscles and changes in body composition. Musculoskeletal symptoms and loss of physical function may be associated with sarcopenia, which is characterized by muscle loss. This study aimed to investigate the prevalence of sarcopenia and its association with RA in a Korean population. We analyzed nationwide data from the Korea National Health and Nutrition Examination Survey of 7389 men and 9798 women. Binomial logistic regression models were used to calculate the odds ratios (ORs) and 95% confidence intervals (CIs) for sarcopenia prevalence in participants with RA. The prevalence of sarcopenia was 23.0% in men, 25.0% in women, 61.5% in men with RA, 32.3% in women with RA, 22.8% in men without RA, and 24.9% in women without RA. After adjusting for potential confounding variables, the prevalence of sarcopenia was higher in men with RA than in men without RA (OR, 3.11; 95% CI, 1.29–7.46), but this difference was not observed in women. In subgroup analysis which was stratified by age (age under 40, age between 40 and 59, age over 60), the OR for sarcopenia was higher in men with age over 60 years (OR, 4.12; 95% CI, 1.48–11.44) and women with age between 40 and 59 (OR, 2.29; 95% CI, 1.05–5.00). The prevalence of sarcopenia was higher in Korean men with RA and women with RA in middle age, suggesting the management of muscle loss will be needed, especially in Koreans with RA.

## 1. Introduction

Sarcopenia was first defined in 1989 as an age-related decrease in muscle mass [1]. It is related to multiple factors and associated with chronic inflammation, such as oxidative stress and increased levels of pro-inflammatory cytokines, including tumor necrosis factor-alpha (TNF-α), interleukin (IL)-6, and IL-1 [2], which are the major cytokines affecting the development and progression of inflammatory arthritis, including rheumatoid arthritis (RA) [3]. RA is a chronic inflammatory disease characterized by synovitis of several joints that causes deformity and destruction of the joints as it progresses [4]. Musculoskeletal symptoms accompanying RA, including pain, swelling, and stiffness, cause a decrease in physical activity, leading to muscle atrophy and weakness [5]. When the levels of inflammatory cytokines associated with RA increase, the body cell mass decreases, muscle weakness and wasting occur, and the body composition changes [6,7].

There have been several studies on the association between RA and sarcopenia [8,9] and the prevalence of sarcopenia in RA [10,11,12]. RA causes pain and deformity of the joints, and sarcopenia causes fatigue and weakening of muscle strength, resulting in a decrease in daily life functioning; thus, both diseases lower quality of life [4,13]. In the elderly, the prevalence of sarcopenia in Korea is >10% in both men and women [14]. Furthermore, the population over 65 years of age is expected to gradually increase, suggesting that the prevalence of sarcopenia will also gradually increase. The prevalence and incidence of RA in Korea are also increasing, and the prevalence rate is high among 60 and 70-year-olds [15,16]. To the best of our knowledge, no study has been conducted on the relationship between RA and sarcopenia in Koreans. Therefore, the purpose of our study was to determine whether the prevalence of sarcopenia is higher in Koreans with RA and to investigate whether there is a significant association between RA and sarcopenia.

## 2. Materials and Methods

### 2.1. Study Design and Setting

This nationwide cross-sectional study used data from the Korea National Health and Nutrition Examination Survey (KNHANES) IV and V from 2008 to 2011. KNHANES is a nationally representative survey administered to a sample of the non-institutionalized civilian population of Korea. Households were randomly selected for participation and sampled using multistage stratifications based on geographical areas.

### 2.2. Participants

In the 2008–2011 KNHANES, 37,753 participants completed RA-related questionnaires. We excluded 16,450 individuals who did not undergo dual-energy X-ray absorptiometry (DXA) and 4116 individuals who did not complete the health survey. Thus, the final sample size of our study was 17,187 participants, of which 7389 were men, and 9798 were women (Figure 1).

### 2.3. Main Variables and Covariates

Participants with RA were identified on the basis of a physician’s diagnosis and current treatment for RA based on participant self-reporting. Otherwise, participants were identified as not having RA. Sarcopenia was defined as appendicular skeletal muscle mass (ASM)/body weight (wt) more than one standard deviation below the mean of a sample of men and women aged 20–40 years [17]. In this study, the cutoff values of ASM/wt were 30.111% and 23.717% in men and women, respectively. ASM was measured in kilograms using DXA (Lunar Corp., Madison, WI, USA) and defined as the sum of lean soft tissue masses of the arms and legs using the method of Heymsfield et al. [18]. Sex, age, obesity, current menstrual status, smoking status, diabetes mellitus (DM), hypertension (HTN), dyslipidemia, alcohol consumption, household income, and education level were considered potential confounding variables affecting the prevalence of RA and sarcopenia. The data were stratified by sex, and men and women were analyzed separately. According to the Korea Center for Disease Control and Prevention (KCDC) guidelines, we included participants whose body mass index (BMI) was <18.5 kg/m^2^ in the underweight group, ≥18.5 kg/m^2^ and <25 kg/m^2^ in the normal weight group, and ≥25 kg/m^2^ in the overweight group. Menopause was defined as self-reported cessation of menstruation for 12 months following the last menstrual period. HTN was defined as an average systolic blood pressure (SBP) of ≥140 mmHg, diastolic blood pressure (DBP) of ≥90 mmHg, or prescription of antihypertensive drugs. Pre-HTN was defined as SBP ≥ 120 mmHg or DBP ≥ 80 mmHg without HTN. DM was defined as a fasting plasma glucose level ≥ 126 mg/dL, with a diagnosis of DM made by a clinician or the prescription of an oral hypoglycemic agent or insulin. Impaired fasting glucose was defined as a fasting plasma glucose level of ≥100 mg/dL and ≤126 mg/dL without a diagnosis of DM. The diagnosis of dyslipidemia was based on the following: total cholesterol level ≥ 200 mg/dL, triglyceride level ≥ 150 mg/dL, high-density lipoprotein cholesterol level < 40 mg/dL in men and <50 mg/dL in women, or current use of any anti-dyslipidemia drugs to control blood lipid concentrations. Alcohol consumption status was defined by the amount of alcohol consumed. Heavy drinkers were defined as those who consumed an average of ≥7 units of alcohol among men and ≥5 units among women ≥2 days/week; moderate drinkers were defined as those who consumed more than one glass of alcohol per month over the past year, and non-drinkers were defined as those who had never consumed alcohol or had consumed less than one glass of alcohol per month over the past year. According to their smoking status, participants were categorized as never smokers, past smokers, or current smokers. Household income levels were divided into quartiles based on monthly income, and education level was classified as primary school or lower, middle school, high school, and university or higher.

### 2.4. Subgroup Analysis

We categorized the participants into 3 groups based on age: participants with age < 40 years, 40 ≤ age < 59 years, and age ≥ 60 years. The prevalence of RA and sarcopenia was calculated in each group, and the odds ratio (OR) for sarcopenia was also analyzed.

### 2.5. Statistical Analysis

The prevalence of RA was described using descriptive statistics. Chi-square tests were used to examine the differences in categorical variables between the RA and non-RA groups, and Student’s *t*-tests were used to compare continuous variables. ORs and 95% confidence intervals (CIs) for sarcopenia according to RA were calculated using logistic regression models. Four logistic regression models were used to assess the association between RA and sarcopenia. Model I was adjusted for age, BMI, and menopause in women. Model II was adjusted for age, BMI, and current menstrual status in women; DM; HTN; and dyslipidemia. Model III was adjusted for age, BMI, and menopause in women; DM; HTN; dyslipidemia; smoking status; and alcohol consumption. Model IV was adjusted for age, BMI, and menopause in women; DM; HTN; dyslipidemia; smoking status; alcohol consumption; household income; and education levels. IBM SPSS statistical software version 23.0 (IBP Corp., Armonk, NY, USA) was used for statistical analyses, and a *p* value of <0.05 was considered statistically significant. In subgroup analysis, OR for sarcopenia was analyzed using Model IV, which was adjusted for age, BMI, and menopause in women; DM; HTN; dyslipidemia; smoking status; alcohol consumption; household income; and education levels.

## 3. Results

### 3.1. Baseline Characteristics

Table 1 shows the demographics of the study sample according to the presence of RA and stratified by sex. In both men and women, the mean age and prevalence of HTN and DM were higher, and household income and education levels were relatively lower in the RA group than in the non-RA group. Among men, smokers and alcohol drinkers were more common in the non-RA group than in the RA group. In women, pre-menopausal status and alcohol consumption were more common in the non-RA group than in the RA group.

The mean of ASM/wt was 29.265% ± 3.018 in men with RA and 32.224% ± 2.862 in men without RA. The mean of ASM/wt was 25.065% ± 2.921 in women with RA and 25.428% ± 2.562 in women without RA. The prevalence of RA among participants in this study was 0.9% (men, 0.4%; women, 1.3%). The prevalence of sarcopenia was 24.1% in all participants (men, 23.0%; women, 25.0%). The prevalence of sarcopenia in the RA group was 37.2% (men, 61.5%; women, 32.3%), which was higher than that in the non-RA group (men, 22.8%; women, 24.9%).

### 3.2. Association between RA and Sarcopenia

In men, the OR of sarcopenia was significantly higher in the RA group than in the non-RA group in the crude model (OR 5.41, 95% CI 2.45–11.95, *p* = 0.001). When covariates were adjusted sequentially, the OR of sarcopenia was significantly higher in men with RA in all models (Model I, OR 2.95, 95% CI 1.25–6.97, *p* = 0.014; Model II, OR 3.15, 95% CI 1.25–6.97, *p* = 0.010; Model III, OR 3.02, 95% CI 1.26–7.20, *p* = 0.013; Model IV, OR 3.11, 95% CI 1.29–7.46, *p* = 0.011). There were no significant differences in the ORs of sarcopenia between the RA and non-RA groups in any of the models for women (crude model, OR 1.44, 95% CI 0.99–2.08, *p* = 0.054, Model I, OR 1.32, 95% CI 0.088–1.97, *p* = 0.183, Model II, OR 1.32, 95% CI 1.32, 95% CI 0.88–1.98, *p* = 0.174; Model III, OR 1.32, 95% CI 0.88–1.97, *p* = 0.184; Model IV, OR 1.36, 95% CI 0.91–2.04, *p* = 0.139). The ORs were calculated after adjusting for age, current menstrual status, chronic diseases, and other lifestyle factors, including socioeconomic status (Table 2).

### 3.3. Association between RA and Sarcopenia in Subgroup Analysis

The baseline characteristics of each subgroup according to the presence of RA and sex were shown in Appendix A. The prevalence of RA among participants with age < 40 years was 0.2% (men, 0.04%; women, 0.3%) and the prevalence of sarcopenia was 15.1% (men, 15.3%; women, 15.0%). The prevalence of sarcopenia in the RA group was 16.7% (men, 100%; women, 9.1%), while that in the non-RA group was 15.1% (men, 15.3%; women, 15.0%). In the group with age ≥ 40 and age < 59 years, the prevalence of RA was 0.5% (men, 0.1%; women, 0.8%), and the prevalence of sarcopenia was 23.7% (men, 21.1%; women, 25.7%). The prevalence of sarcopenia in RA was 40.0% (men, 25.0%; women, 41.9%), and that in the non-RA group was 23.5% (men, 21.1%; women, 25.5%). In the group with age ≥ 60 years, the prevalence of RA was 2.2% (men, 1.0%; women, 3.1%), and the prevalence of sarcopenia was 34.7% (men, 33.6%; women, 35.5%). The prevalence of sarcopenia in RA was 38.5% (men, 66.7%; women, 31.8%), while that in the non-RA group was 34.6% (men, 33.3%; women, 35.6%) (Table 3).

Using model IV, the ORs of sarcopenia were calculated in each age group. In men with age < 40 years, the number of RA was only 1, and the OR of sarcopenia was not calculated. In men with 40 ≤ Age < 59 years, the OR of sarcopenia in RA was not different (OR 0.79, 95% CI 0.08–7.97, *p* = 0.839). However, in men with age > 60 years, the OR of sarcopenia was significantly higher than in the non-RA group (OR 4.12, 95% CI 1.48–11.44, *p* = 0.007). In women with age < 40 years, the OR of sarcopenia was not different in the RA and non-RA groups (OR 0.56, 95% CI 0.07–4.82, *p* = 0.598). In women with 40 ≤ Age < 59 years, the OR for sarcopenia was significantly high in the RA group (OR 2.29, 95% CI 1.05–5.00, *p* = 0.038). On the other hand, in women with age > 60 years, the OR of sarcopenia was not different (OR 1.30, 95% CI 0.78–2.16, *p* = 0.317) (Table 4).

## 4. Discussion

In this study, the prevalence of RA was 0.9% and was approximately three times higher in women than in men. The prevalence of sarcopenia in participants with RA was 37.2%, significantly higher than that in participants without RA (24.0%). A previous meta-analysis reported the prevalence of sarcopenia in RA as 35.1%, which is similar to that observed in this study [19]. However, in a previous meta-analysis, the prevalence was 34.2% in men and 33.1% in women, whereas, in our study, the prevalence was approximately two times higher in men than in women (61.5% in men vs. 32.3% in women). This suggests that, in Korea, men with RA may need more management to prevent muscle loss or weakness than women. The high prevalence of sarcopenia in the RA group may be due to age differences. In both men and women, the participants in the RA group were older than those in the non-RA group; in men, the age difference between the two groups was approximately 20 years, which was greater than that observed in women. The differences in the prevalence of sarcopenia between men and women in RA may have been large because muscle mass begins to decrease around the age of 40 years in men and after the age of 55 years in women [20].

In addition to age, various other factors influence the development of sarcopenia with RA. In men, the OR for sarcopenia in the RA group was significantly high in all models. In women, there was no significant difference in the OR of sarcopenia between the RA and non-RA groups in any model. Menopause leads to a change in endocrine functions, resulting in a decrease in muscle mass [21], and this study showed a significant difference in the menopause status between the RA and non-RA groups. However, even after adjusting for menopausal status, no relationship was found between the presence of RA and sarcopenia. Although the prevalence of sarcopenia is higher in women than in men, decreases in muscle mass, strength, and physical function are more pronounced in men than in women [22]. Thus, the decrease in muscle mass and strength associated with impaired physical function due to RA can be more pronounced in men than in women. These results indicate that appropriate management is essential for preventing sarcopenia in men with RA. In subgroup analysis, the ORs of sarcopenia were not different in ages < 40 years and age between 40 and 59 in men. The small number of participants with RA in each group, only one and three, respectively, may have influenced the result. The OR of sarcopenia was still significantly higher in the participants aged 60 years or older, who accounted for the majority of RA participants. In women, the OR of sarcopenia was significant in RA participants aged between 40 and 59 but not in RA participants aged 60 years or older. In a recent study, the mean age of menopause in Korea was 49.9 years [23]. Since the group aged between 40 and 59 included participants who were premenopausal or had not been menopausal for long, the OR of sarcopenia might have been high in the RA group, as it was in men. Whereas in the group aged 60 years or older, it can be assumed that the ORs of sarcopenia between the RA and non-RA groups may not have been significantly different due to the decrease in muscle mass caused by menopause in the non-RA group. There are several criteria for defining sarcopenia; it can be evaluated using muscle mass but also by muscle strength and physical function [24]. However, since the method for evaluating muscle strength and physical function has not been clearly defined, the error range may be large. In this study, sarcopenia was defined by evaluating muscle mass only. There are methods that use height (ASM/ht^2^), weight (ASM/wt), or BMI (ASM/BMI) to define sarcopenia based on ASM. The prevalence of sarcopenia in Korean men measured using ASM/wt was similar to that measured using ASM/ht^2^ and higher than that measured using ASM/BMI. In Korean women, the prevalence of sarcopenia as measured by ASM/ht^2^ was <10% and that measured by ASM/wt was higher than that measured by ASM/ht^2^ but lower than that measured by ASM/BMI [25]. In women, the prevalence of sarcopenia was low when ASM/ht^2^ was applied, whereas it tended to be high in both men and women when ASM/BMI was applied. Thus, sarcopenia was defined as ASM/wt in the present study.

Older age is the most common cause of sarcopenia. However, chronic inflammation has also been associated with sarcopenia. Thus, in 2018, the European Working Group on Sarcopenia in Older People defined secondary sarcopenia, which is associated with chronic inflammation, immobility, or malnutrition [26]. The main pathology of RA is a chronic inflammation of the synovium, in which pro-inflammatory cytokines, including TNF-α, IL-6, and IL-1β, play a key role. Studies on the bidirectional pathological relationship between RA and sarcopenia are increasing [27]. Sarcopenia is also associated with elevated levels of TNF-α, IL-6, and C-reactive protein, suggesting that appropriate treatment for RA may also reduce the development of sarcopenia [11,28].

This study is the first to investigate the prevalence of sarcopenia in people with RA and the association between sarcopenia and RA in the Korean population. As we used authoritative nationwide data, this study represents the prevalence and association between sarcopenia and RA in the general Korean population. The participants underwent DXA, which is a quantitative diagnostic modality for measuring muscle mass and is superior to bioelectric impedance analysis [29]. Furthermore, our analyses were adjusted for important confounding factors, such as age, current menstrual status, chronic diseases, and food indulgence, which affect the prevalence of RA and sarcopenia. Several studies reported on the association between RA and sarcopenia [8,19]. The risk factors of sarcopenia in these studies were age, longer disease duration, malnutrition, DAS28, C-reactive protein, and rheumatoid factor seropositivity. Other studies reported on the relationship between menopause and sarcopenia [30,31]. However, there was no study that conducted an age-stratified analysis of the association between RA and sarcopenia. In our study, there was no difference in the OR of sarcopenia between the RA and non-RA groups in women initially. After conducting a subgroup analysis, we found that there was a difference in the OR of sarcopenia between women with aged between 40 and 59 and women aged over 60. It can be assumed that menopause may have influenced this result. Therefore, further research is necessary to investigate this possibility.

Our study has some limitations. First, since this was a cross-sectional study, it could not show a causal relationship between RA and sarcopenia. Second, the data may have been influenced by systematic errors in responses from individuals, which may have led to non-differential misclassification as the definition of the main variable was based on a self-report survey. The definition of RA was also determined by self-report, and data on sero-positivity, such as rheumatoid factor and anti-citrullinated protein/peptide antibody levels, were not available. Patients with other diagnoses, such as lumbar pain, osteoarthritis, or gout, may have been included in RA. Third, the duration, activity, and severity of RA may have affected the development of sarcopenia [27]; however, these were not investigated in this study. Finally, sarcopenia was defined by evaluating muscle mass only and not physical function, activity, or muscle strength. DXA measurement can overestimate muscle mass in participants with obesity or extracellular fluid accumulation [32].

## 5. Conclusions

The prevalence of sarcopenia was higher in Korean men and middle-aged women with RA than in those without RA. Therefore, appropriate monitoring and management are required to reduce the development of sarcopenia patients with RA.

## Figures and Tables

**Figure 1 healthcare-11-01401-f001:**
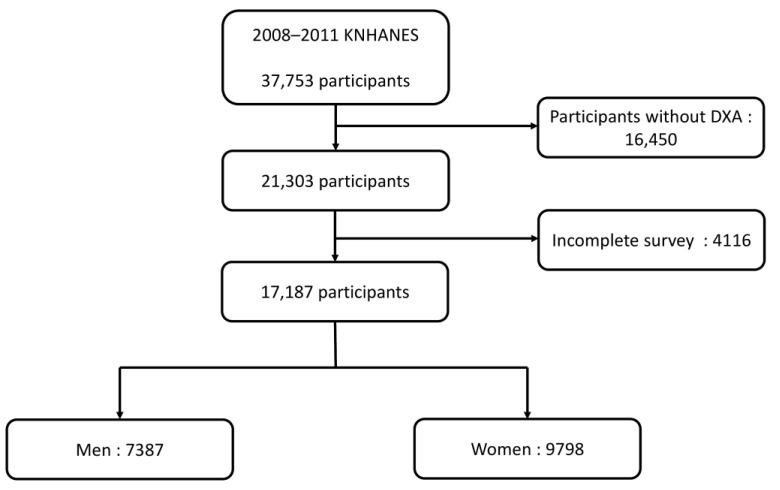
Flowchart of participants’ selection. KNHANES, the Korean National Health and Nutrition Examination Survey; DXA, dual-energy X-ray absorptiometry.

**Table 1 healthcare-11-01401-t001:** Baseline characteristics of the participants with and without rheumatoid arthritis.

	Men (*n* = 7389)					Women (*n* = 9798)				
	RA (*n* = 26)		Non-RA (*n* = 7363)			RA (*n* = 130)		Non-RA (*n* = 9668)		
	*n*	%	*n*	%	*p*	*n*	%	*n*	%	*p*
**Age (in years)**	67.54 ± 12.281		48.95 ± 16.012		0.001	61.92 ± 13.273		48.62 ± 16.071		0.001
**Obesity**					0.087					0.394
Underweight	0	0.0	241	3.3		7	5.4	534	5.5	
Normal weight	13	50.0	4534	61.6		91	70.0	6375	65.9	
Overweight	13	50.0	2588	35.1		32	24.6	2759	28.5	
**Menopause**					-					0.001
Yes	-	-	-	-		105	80.8	4099	42.4	
No	-	-	-	-		25	19.2	5569	57.6	
**Hypertension**					0.041					0.001
Normal	8	30.8	2484	33.7		32	24.6	5029	52.0	
Pre-hypertension	3	11.5	2220	30.2		35	26.9	1967	20.3	
Hypertension	15	57.7	2659	36.1		63	48.5	2672	27.6	
**Diabetes mellitus**					0.000					0.036
Normal	15	57.7	4867	66.1		90	69.2	7386	76.4	
Impaired fasting glucose	2	7.7	1683	22.9		21	16.2	1464	15.1	
Diabetes mellitus	9	34.6	813	11.0		19	14.6	818	8.5	
**Dyslipidemia**					0.604					0.051
Normal	10	38.5	2477	33.6		25	19.2	2597	26.9	
Dyslipidemia	16	61.5	4886	66.4		105	80.8	7071	73.1	
**Alcohol consumption**					0.001					0.000
None	16	61.5	1857	25.2		101	77.7	5881	60.8	
Moderate	8	30.8	3981	54.1		28	21.5	3378	34.9	
Heavy	2	7.7	1525	20.7		1	0.8	409	4.2	
**Smoking status**					0.039					0.779
Never	4	15.4	1406	19.1		113	86.9	8592	88.9	
Past	14	53.8	2803	38.1		8	6.2	518	5.4	
Current	8	30.8	3154	42.8		9	6.9	558	5.8	
**Household income**					0.001					0.001
Lowest	15	57.7	1281	17.4		57	43.8	2020	20.9	
Lower middle	5	19.2	1848	25.1		30	23.1	2451	25.4	
Upper middle	1	3.8	2141	29.1		26	20.0	2631	27.2	
Highest	5	19.2	2092	28.4		17	13.1	2566	26.5	
**Education**					0.001					0.000
Primary school or lower	13	50.0	1315	17.9		89	68.5	3051	31.6	
Middle school	5	19.2	913	12.4		19	14.6	1008	10.4	
High school	6	23.1	2658	36.1		13	10.0	3238	33.5	
University or higher	2	7.7	2477	33.6		9	6.9	2371	24.5	
**Sarcopenia**					0.001					0.053
Absent	10	38.5	5683	77.2		88	67.7	7260	75.1	
Present	16	61.5	1680	22.8		42	32.3	2408	24.9	

RA = rheumatoid arthritis. Data are given as mean ± standard deviation or the number with percentage.

**Table 2 healthcare-11-01401-t002:** Odds ratios and 95% confidence intervals for sarcopenia in patients with rheumatoid arthritis.

	Men			Women		
	OR	95% CI	*p*	OR	95% CI	*p*
Crude	5.41	2.45–11.95	0.001	1.44	0.99–2.08	0.054
Model I	2.95	1.25–6.97	0.014	1.32	0.88–1.97	0.183
Model II	3.15	1.25–6.97	0.010	1.32	0.88–1.98	0.174
Model III	3.02	1.26–7.20	0.013	1.32	0.88–1.97	0.184
Model IV	3.11	1.29–7.46	0.011	1.36	0.91–2.04	0.139

Model I has been adjusted for age, BMI, and current menstrual status especially in women. Model II has been adjusted for Model I + DM, HTN, and dyslipidemia. Model III has been adjusted for Model II + alcohol consumption and smoking status. Model IV has been adjusted for Model III + household income and education. OR, odds ratio; CI, confidence interval; BMI, body mass index; DM, diabetes mellitus; HTN, hypertension.

**Table 3 healthcare-11-01401-t003:** The prevalence of rheumatoid arthritis and sarcopenia according to age groups.

	Men (*n* = 2399)					Women (*n* = 3239)				
Age < 40 Years	RA (*n* = 1)		Non-RA (*n* = 2398)			RA (*n* = 11)		Non-RA (*n* = 3228)		
	*n*	%	*n*	%	*p*	*n*	%	*n*	%	*p*
**Sarcopenia**					0.153					0.713
Absent	0	0	2031	84.7%		10	90.9%	2743	85.0%	
Present	1	100.0%	367	15.3%		1	9.1%	485	15.0%	
	**Men (*n* = 2802)**					**Women (*n* = 3760)**				
**40 ≤ Age < 59 years**	**RA (*n* = 4)**		**Non-RA (*n* = 2798)**			**RA (*n* = 31)**		**Non-RA (*n* = 3729)**		
	** *n* **	**%**	** *n* **	**%**	** *p* **	** *n* **	**%**	** *n* **	**%**	** *p* **
**Sarcopenia**					0.849					0.039
Absent	3	75.0%	2207	78.9%		18	58.1%	2772	74.3%	
Present	1	25.0%	591	21.1%		13	41.9%	957	25.7%	
	**Men (*n* = 2188)**					**Women (*n* = 2799)**				
**Age ≥ 60 years**	**RA (*n* = 21)**		**Non-RA (*n* = 2167)**			**RA (*n* = 88)**		**Non-RA (*n* = 2711)**		
	** *n* **	**%**	** *n* **	**%**	** *p* **	** *n* **	**%**	** *n* **	**%**	** *p* **
**Sarcopenia**					0.001					0.462
Absent	7	33.3%	1445	66.7%		60	68.2%	1745	64.4%	
Present	14	66.7%	722	33.3%		28	31.8%	966	35.6%	

RA = rheumatoid arthritis. Data are given as the number with percentage.

**Table 4 healthcare-11-01401-t004:** Odds ratios and 95% confidence intervals for sarcopenia in patients with rheumatoid arthritis according to age groups.

	Men			Women		
	OR	95% CI	*p*	OR	95% CI	*p*
All age	3.11	1.29–7.46	0.011	1.36	0.91–2.04	0.139
Age < 40 years	-			0.56	0.07–4.82	0.598
40 ≤ Age < 59 years	0.79	0.08–7.97	0.839	2.29	1.05–5.00	0.038
Age ≥ 60 years	4.12	1.48–11.44	0.007	1.30	0.78–2.16	0.317

Adjusted for age, BMI, current menstrual status in women, hypertension, diabetes mellitus, dyslipidemia, alcohol consumption, smoking status, household income, and education. OR, odds ratio; CI, confidence interval.

## Data Availability

All National Health and Nutrition Examination Survey files are available from the Korea Centers for Disease Control and Prevention database (URL https://knhanes.kdca.go.kr/knhanes/sub03/sub03_02_05.do, accessed on 11 March 2023).

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
