# Peer review of "The High Prevalence of Sarcopenia in Rheumatoid Arthritis in the Korean Population: A Nationwide Cross-Sectional Study"

_healthcare, 2023, doi:10.3390/healthcare11101401_

Round 1
Reviewer 1 Report
Dear authors,
I have reviewed your manuscript and have several suggestions for improvement:
Abstract: Please correct the term "musculoskeletal" symptoms in the first line.
Introduction: The reference for the information that "Sarcopenia was first defined in 1989" was not the original source (Rosenberg, I.H. Summary comments. Am. J. Clin. Nutr. 1989, 50, 1231–1233). Please correct this.
Methods: Study design: Please provide references for KNHANES.
Results: Please provide the values of ASM/wt for subjects with and without RA, as well as the cutoff points for considering the occurrence of sarcopenia.
Discussion: I recommend writing the paragraph on study limitations before drawing conclusions. It would be interesting to reinforce that the RA diagnosis was self-reported, and that patients with other diagnoses, such as lumbar pain, osteoarthritis (especially hip), or gout, may have been included. The absence of a clinical criterion for sarcopenia, such as grip strength or physical performance, is also an important limitation. It would be beneficial to discuss the potential impact of these limitations on the study findings.
RA tends to be more common in women than in men, and disability in rheumatoid women tends to be worse or similar to that of men with this disease. It would be interesting to note or compare your results with those of other authors. I have not found any studies that observed gender as a cause of sarcopenia in RA. Most studies report disease-related factors such as disease activity, disease duration, disability, PCR, rheumatoid factor, anti-CCP, and hip osteoarthritis. This topic could be further discussed.
Sincerely,
Author Response
Dear reviewer,
Thank you for your detailed review. Here are our revisions for each points
- Abstract: Please correct the term "musculoskeletal" symptoms in the first line.
- We have corrected the typo.
- Introduction: The reference for the information that "Sarcopenia was first defined in 1989" was not the original source (Rosenberg, I.H. Summary comments. Am. J. Clin. Nutr. 1989, 50, 1231–1233). Please correct this.
- We also corrected the appropriate reference.
- Methods: Study design: Please provide references for KNHANES.
- The reference is already listed in the “data availability statement”, at the end of manuscript.
- Results: Please provide the values of ASM/wt for subjects with and without RA, as well as the cutoff points for considering the occurrence of sarcopenia.
- Values of ASM/wt for participants with and without RA was added in the result section (line 129-130)
- Discussion: I recommend writing the paragraph on study limitations before drawing conclusions. It would be interesting to reinforce that the RA diagnosis was self-reported, and that patients with other diagnoses, such as lumbar pain, osteoarthritis (especially hip), or gout, may have been included. The absence of a clinical criterion for sarcopenia, such as grip strength or physical performance, is also an important limitation. It would be beneficial to discuss the potential impact of these limitations on the study findings.
- We already mentioned about the limitation of this study in discussion. We have made some additional changes based on your review.
- RA tends to be more common in women than in men, and disability in rheumatoid women tends to be worse or similar to that of men with this disease. It would be interesting to note or compare your results with those of other authors. I have not found any studies that observed gender as a cause of sarcopenia in RA. Most studies report disease-related factors such as disease activity, disease duration, disability, PCR, rheumatoid factor, anti-CCP, and hip osteoarthritis. This topic could be further discussed.
- In subgroup analysis, we found difference in the risk of sarcopenia in middle-aged women. We compared our result with other studies in terms of risk factors, gender differences, and impact of menopause in discussion section.
Reviewer 2 Report
The study written by Kim, et al is well written and the results have a clinical and epidemiological significance in people with rheumatoid arthritis. However, I think there are critical concerns in the interpretation of the study results. I believe revision must prevent misinterpretation of the results.
1. In the Abstract and Results, the authors should consider the overestimation of an odds ratio when it is interpreted as a prevalence ratio because of the high prevalence of sarcopenia. I cannot agree with the expression as “three-fold higher” and “three-time” prevalence in the concluding sentence of the abstract and the manuscript, respectively, if they were referred to the adjusted odds ratio of 3.11. I think, in this study, the odds ratios cannot be interpreted as an assessment of prevalence because of the frequent outcome, but rather as whether there is an association and in what direction.
[Ref. Gnardellis C, et al. Overestimation of Relative Risk and Prevalence Ratio: Misuse of Logistic Modeling. Diagnostics (Basel). 2022;12(11):2851. doi: 10.3390/diagnostics12112851.].
2. In the Materials and Methods, 2.2 participants, a lot of participants were excluded from the overall participants in KNHANES. So, the authors could consider using a flow diagram of the participants' selection or report numbers of individuals at each stage of the study including.
3. In the present study, I cannot find any sensitivity analyses or subgroup analyses. Given the study topic, at least I think results stratified by age group would be needed.
4. Although the present study is a cross-sectional study, it is easier to understand if the title is described in alignment with the exposure to outcome direction.
Author Response
Dear reviewer,
Thank you for your important review. Here are our revisions for each points
- In the Abstract and Results, the authors should consider the overestimation of an odds ratio when it is interpreted as a prevalence ratio because of the high prevalence of sarcopenia. I cannot agree with the expression as “three-fold higher” and “three-time” prevalence in the concluding sentence of the abstract and the manuscript, respectively, if they were referred to the adjusted odds ratio of 3.11. I think, in this study, the odds ratios cannot be interpreted as an assessment of prevalence because of the frequent outcome, but rather as whether there is an association and in what direction.
[Ref. Gnardellis C, et al. Overestimation of Relative Risk and Prevalence Ratio: Misuse of Logistic Modeling. Diagnostics (Basel). 2022;12(11):2851. doi: 10.3390/diagnostics12112851.].
- We agree with your opinion. We change the expressions “three-fold higher” and “three time” in manuscript. “The prevalence of sarcopenia was more than three-fold higher” in abstract, line 25 has been changed to “The prevalence of sarcopenia was higher”. And other expressions used in discussion such as “approximately three times higher” (line 190), “OR for the prevalence of sarcopenia in the RA group was more then three times higher than that in the non-RA group” (line 204) has been modified to the expression “risk of the sarcopenia”.
- In the Materials and Methods, 2.2 participants, a lot of participants were excluded from the overall participants in KNHANES. So, the authors could consider using a flow diagram of the participants' selection or report numbers of individuals at each stage of the study including.
- We included both the flowchart of participants’ selection (Figure 1) and number of individuals at each stage in section 2.2 participants.
- In the present study, I cannot find any sensitivity analyses or subgroup analyses. Given the study topic, at least I think results stratified by age group would be needed.
- This was a very important point, we think. Subgroup analysis was performed and the result of subgroup analysis was included. In subgroup analysis, women in middle-age showed risk of sarcopenia. We also added discussions about the result.
- Although the present study is a cross-sectional study, it is easier to understand if the title is described in alignment with the exposure to outcome direction.
- Title was changed to show the conclusion of the study.
Round 2
Reviewer 1 Report
To the authors:
I was pleased to see that you have compared subjects with rheumatoid arthritis to those without, by age. However, this change in methodology was only mentioned in the results section and not in the methods section.
I have made some suggestions, and I hope it could help you:
Abstract: The abstract should include the new methodology and results that separated patients by age group (<40 years, 40-59 years, >60 years).
Methods: You need to describe this analysis of patients by age group in the methods of your study, not just in the results. Also, describe how the statistical analysis was performed in statistical analysis.
Results:
Major comments: I recommend you to describe patients baseline characteristics according to age group. This is especially important in women group, especially regarding menopause since you have discussed that menopause may explain differences in sarcopenia from women with and without rheumatoid arthritis.
Minor: Include the tests used in the analyses in the footnotes of the tables.
Discussion:
Major comments: On page 7, second paragraph, it is written "After conducting subgroup analysis, we found that there was a difference in the risk of sarcopenia between pre and postmenopausal women." I could not find this data in the results of your study. In fact, since you analyzed by age group and not by pre and post-menopause, you should stick to this data. The issue of menopause may be stated as an assumption.
Author Response
Dear reviewer,
Thank you for your kind review and suggestions. We made revisions for each points.
- Abstract: The abstract should include the new methodology and results that separated patients by age group (<40 years, 40-59 years, >60 years).
- We modified the abstract. The definition of subgroup and result of subgroup analysis was included in abstract.
- Methods: You need to describe this analysis of patients by age group in the methods of your study, not just in the results. Also, describe how the statistical analysis was performed in statistical analysis.
- We made a new section of subgroup analysis in methods and described the stratifying method. And we also described the statistical analysis in statistical analysis section. Thank you for pointing out the part we missed.
- Results:
Major comments: I recommend you to describe patients baseline characteristics according to age group. This is especially important in women group, especially regarding menopause since you have discussed that menopause may explain differences in sarcopenia from women with and without rheumatoid arthritis.
- We included supplement tables (table S1-S3) which showed the baseline characteristics of each groups.
- Minor: Include the tests used in the analyses in the footnotes of the tables.
- We included the tests used in analyses in the footnotes of the tables.
- Discussion:
Major comments: On page 7, second paragraph, it is written "After conducting subgroup analysis, we found that there was a difference in the risk of sarcopenia between pre and postmenopausal women." I could not find this data in the results of your study. In fact, since you analyzed by age group and not by pre and post-menopause, you should stick to this data. The issue of menopause may be stated as an assumption
- We agree with your opinion. The issue of menopause is just an assumption. So we modified the discussion and conclusion. On discussion, The expression “ a difference in the risk of sarcopenia between pre and post menopausal women” is changed to “a difference in the OR of sarcopenia between women with aged between 40 and 59 and women with aged over 60”. Other part in discussion and conclusions were also modified.
Reviewer 2 Report
I could see the manuscript has been improved after the revision. But there remain concerns in reporting results.
1. Because the present study is a cross-sectional study, the term “the risk” cannot be used in the result reporting in the Results and Discussion sections. I think it should be described as “the odds ratio”.
2. In lines 263-264, discussion section, the authors stated “After conducting subgroup analysis, we found that there was a difference in the risk of sarcopenia between pre and postmenopausal women.” I think the age-stratified analysis cannot determine a difference between pre and post-menopausal though the authors reported the mean age of menopause (line 220). Also, I cannot agree with the description “premenopausal women” in line 280, conclusion section, due to the same reason.
Author Response
Dear reviewer,
Thank you for your comments. We made revisions for each points.
- Because the present study is a cross-sectional study, the term “the risk” cannot be used in the result reporting in the Results and Discussion sections. I think it should be described as “the odds ratio”.
- The terms “the risk” are all changed to “the odds ratio” in results and discussion session. We agree with your opinion.
- In lines 263-264, discussion section, the authors stated “After conducting subgroup analysis, we found that there was a difference in the risk of sarcopenia between pre and postmenopausal women.” I think the age-stratified analysis cannot determine a difference between pre and post-menopausal though the authors reported the mean age of menopause (line 220). Also, I cannot agree with the description “premenopausal women” in line 280, conclusion section, due to the same reason.
- We totally agree with your opinion. We change the expression “ a difference in the risk of sarcopenia between pre and post menopausal women” to “a difference in the OR of sarcopenia between women with aged between 40 and 59 and women with aged over 60”. Other part in discussion and conclusions were also modified.
Round 3
Reviewer 1 Report
Dear authors,
Thank you for considering my suggestions.
I have reviewed the modifications made to the abstract, methods, results, and discussion, and I find them appropriate.
Even though you did not provide the baseline data for the subjects by age group, as I had suggested, I am satisfied that you removed the conclusions regarding menopause from the discussion in the previous version.
Overall, I believe that the modifications made to the manuscript have improved it.
I wish you all the best in your journey.
Reviewer 2 Report
Overall, I think the manuscript has been revised satisfactorily according to the comments in the review reports.